# Bioremediation of Azo Dye Brown 703 by *Pseudomonas aeruginosa*: An Effective Treatment Technique for Dye-Polluted Wastewater

**Asad Ullah Khan** [1]**, Muhammad Zahoor** [2,*]**, Mujaddad Ur Rehman** [1]**, Muhammad Ikram** [3,*]**, Daochen Zhu** [4]**, Muhammad Naveed Umar** [5]**, Riaz Ullah** [6] **and Essam A. Ali** [7]

[1] Department of Microbiology, Abbottabad University of Science and Technology, Havelian, Abbottabad 22500, Pakistan; asad.microbiologist@gmail.com (A.U.K.); mujaddad@aust.edu.pk (M.U.R.)
[2] Department of Biochemistry, University of Malakand, Chakdara 18800, Pakistan
[3] Department of Chemistry, Abdul Wali Khan University Mardan, Mardan 23200, Pakistan
[4] School of Environment and Safety Engineering, Biofuels Institute, Jiangsu University, Zhenjiang 212013, China; dczhucn@hotmail.com
[5] Department of Chemistry, University of Liverpool, Liverpool L69 3BX, UK; m.naveed-umar@liverpool.ac.uk
[6] Department of Pharmacognosy, College of Pharmacy, King Saud University, Riyadh 11451, Saudi Arabia; rullah@ksa.edu.sa
[7] Department of Pharmaceutical Chemistry, College of Pharmacy, King Saud University, Riyadh 11451, Saudi Arabia; esali@ksu.edu.sa
* Correspondence: mohammadzahoorus@yahoo.com (M.Z.); ikrambiochem2014@gmail.com (M.I.)

**Abstract:** Dye-polluted wastewater poses a serious threat to humans', animals' and plants' health, and to avoid these health risks in the future, the treatment of wastewater containing dyes is necessary before its release to environment. Herein, a biological approach is used; the textile azo dye brown 703 is degraded utilizing *Pseudomonas aeruginosa*. The bacterial strain was isolated from textile wastewater dumping sites in Mingora, Swat. The optimization for bacterial degradation was carried out on the nutrient broth medium, which was then subjected to a variety of environmental physicochemical conditions and nutritional source supplementation before being tested. Under micro-aerophilic circumstances, the maximum decolorization and degradation of dye occurred at a 20 ppm dye concentration within 3 days of incubation at a neutral pH and 38 °C. The decrease in the intensity of the absorbance peak in the UV–Vis spectrum was used to measure the extent of decolorization. Initially, 15 bacterial strains were isolated from the textile effluent. Out of these strains, *Pseudomonas aeruginosa* was found to be the most potent degrading bacteria, with a degradation extent of around 71.36% at optimum conditions. The appearance and disappearance of some new peaks in the FT-IR analysis after the degradation of brown 703 showed that the dye was degraded by *Pseudomonas aeruginosa*. The GC–MS analysis performed helped in identifying the degraded compounds of azo dye that were utilized in illustrating the under-study process of brown 703 degradation. The biodegradation brought about by *Pseudomonas aeruginosa* can be employed successfully in the future as an eco-friendly approach with far reaching results.

**Keywords:** biodegradation; bioreactor; textile effluents; *Pseudomonas aeruginosa*; metabolites; water pollution

## 1. Introduction

As worldwide industrialization has evolved in recent years, there has been a tremendous increase in both the production and the use of a wide range of chemicals and dyes in everyday life, which has contributed to the growth of the global economy. A combination of increased industrialization and population growth has resulted in the development of industry as well as transportation and residential sectors all at the same time. All of

these changes are a huge contributing factor to the release of pollutants and toxins into the environment in alarming quantities around the world. Industrialization and urbanization have played an important role in economic development, but they have massively increased the production of wastes, which adversely affect humans, animals and plants [1]. There is a significant amount of wastewater released from the textile industry daily, containing harmful and dangerous chemicals that are difficult to extract from the environment [2]. Furthermore, the introduction of textile waste into the ecosystem limits the amount of light reaching plants growing in water, which has an impact on the photosynthesis of these plants, thus having adverse impacts on the aquatic ecosystem [3].

A variety of contaminants, including dissolved and suspended particles, dyes, and hazardous metals, can be found in the wastewater from the textile industry. Materials used for dyeing objects and clothes have been frequently cited as the most harmful pollutants having a direct impact on humans, animals and plants [4]. According to the reported estimates worldwide, over 10,000 dyes and pigments are now being used in the dyeing and printing industries, and this number is increasing on a daily basis. Most of the synthetic-based dyes utilized in the industrial sector are harmful and carcinogenic to all living things, including humans and other living biota [5]. Worldwide, around $7 \times 10^5$ tons of dyes are manufactured each year, of which the majority are azo dyes [6].

In a wide variety of industrial processes, azo dyes are commonly utilized as coloring agents, and they are particularly significant in the cosmetic, leather, textile, paper, and tanning industries [7]. The use of azo dyes is relatively high compared to other dyes because of their low cost, chemical stability, easy availability, and versatility. In addition to being synthetic dyes made from aromatic components, they also have strong oxidizing properties [8]. When compared to other dyes, the dyeing processes of azo dyes are so exceptional that they are utilized 50% more frequently in textile industries than other dyes [9]. Out of the $7 \times 10^5$ tons of dyes manufactured each year, 10 to 15% of these dyes are released into wastewater, which not only decreases light penetration into deep water but also causes serious health complications [10]. Under the anaerobic conditions that exist in deep aquatic environments, they give rise to different types of amines, some of which are carcinogenic [11]. To lessen the environmental impact of these azo dyes, they must be removed from wastewater effectively.

The brown 703 azo dye used in this research project to test the degradation power of different bacterial strains has the molecular formula $C_{30}H_{20}O_{12}N_8S_2Na_2$, and is a brown-colored dye. This dye is commonly used for dyeing a variety of fabrics, including cotton, nylon, silk, and polyester. Due to the extremely poor fixation rate of brown 703, the amount of dye released by industries is substantially greater than the amount of dye fixed [12]. Many physiochemical methods are employed for the removal of dyes, including filtration, photo degradation, sedimentation, coagulation, adsorption, and electrolysis. However, the application of these methods is limited on an industrial scale due to their inflexibility, high cost, and generation of secondary effluents [13].

The high solubility rate of azo dyes in water is the most significant problem in the remediation of azo dyes [14]. Consequently, wastewater treatment can be accomplished via a variety of approaches, with the biological method being the most natural and successful. This method comprises bacterial degradation [15], fungal degradation [16], phytoremediation [17], and enzymatic degradation [18]. In the biological method, microbes are used to break down complex molecules such as azo dyes and other pollutants into simpler substances. Also, it has been observed that when bacterial strains are used for azo dye decomposition or breakdown, the least amount of sludge is generated. Nowadays, scientists are focused on the bacterial degradation of azo dyes, because bacteria's growth rate is high as compared to other microorganisms. At the same time, it is less expensive than the enzymatic degradation of dyes [19].

Though a number of studies have highlighted the degradation ability of microbes, but no researchers have made an attempt to isolate the metabolites in a pure form to gain insights into the degradation process. Therefore, the specific objectives of this research

study were to isolate different bacterial strains from dye-contaminated water, to determine the degradation efficiency of isolated bacterial strains, and to optimize various physiochemical parameters, including pH, temperature, brown 703 concentration, time (in days), salt concentration, and glucose concentration, which are required for the most effective dye degradation.

## 2. Materials and Methods

### 2.1. Experimental Materials

The azo dye brown 703 nutrient broth was obtained from the Department of Biochemistry, Malakand University, Pakistan. In these investigations, high-purity analytical grade solvents such as methanol and ethyl acetate were utilized.

### 2.2. Brown 703 Dye

The brown 703 molecular formula is $C_{30}H_{20}O_{12}N_8S_2Na_2$, with a molecular weight of 794.63 g/mol. Figure 1 shows its chemical structure.

**Figure 1.** Brown 703 dye chemical structure.

### 2.3. Isolation of Bacterial Strains

Wastewaters from the textile sector, obtained from several locations throughout Swat, Pakistan, were collected and utilized to isolate and purify different bacterial strains for the degradation of brown 703 by using streak and pour plate methods. About a 1 mL sample of wastewater was taken and diluted it up to $10^{-4}$ times in order to isolate bacterial strains. The spread plate technique was used on the last diluted ($10^{-4}$) wastewater sample for bacterial growth. The nutrition agar plate was streaked with diluted contaminated water with the help of a wire loop and then incubated in an incubator at 37 °C for 24 h. Incubation of the nutrient agar plate for 24 h led to the formation of many bacterial colonies. From these colonies, 15 separate bacterial colonies were isolated, purified, and used for the degradation of brown 703.

### 2.4. Brown 703 Degradation Analyses

In order to determine the degrading capacity of the isolated bacterial strains, from each pure culture, bacteria were transferred into a 100 mL flask containing nutrient broth (60 mL) and incubated at 37 °C for 24 h to allow the bacteria to proliferate to produce the inoculum for the subsequent investigational studies. After bacterial growth in the nutrient broth, a small amount of each bacterial culture was introduced separately to each test tube, which contained 20 mL of nutrient broth and 20 ppm of brown 703 dye concentration, and was incubated for 3 days at 37 °C. To analyze the percent degradation of brown 703 by different bacterial strains, each test tube of nutrient broth was centrifuged for 20 min at 5000 rpm to separate the supernatant from the pellet. For the determination of the brown 703 dye degradation rate, the top supernatant was utilized. A UV–Vis spectrophotometer

was used for the measurement of percent bio-decolorization of brown 703 at a wavelength of 471 nm. The following formula was used to calculate brown 703 degradation.

$$\% \text{ Decolorization} = \frac{\text{Initial absorbance} - \text{final absorbance}}{\text{Initial absorbance}} \times 100\% \tag{1}$$

*2.5. Brown 703 Degradation and Effect of Different Physiochemical Parameters*

2.5.1. Effect of Brown 703 Concentrations

A series of experiments were conducted at various concentrations ranging from 20 to 100 parts per million (ppm) in order to determine the efficiency of degradation of the brown 703 by the selected bacteria. The bacterial culture was introduced into each test tube, which contained various concentrations of brown 703. The nutrient broth that was not inoculated with bacteria was used as a control sample. For around three days, the degrading ability of the isolated strain at different dye concentrations was examined. After three days of incubation, each test tube was centrifuged for 20 min at 5000 rpm to separate the pellets and the supernatant. The supernatant of the dye-degraded solution was used to measure the percent degradation using Equation (1).

2.5.2. Effect of pH on Degradation of Brown 703

The pH's impact on brown 703 degradation was analyzed using sterilized nutrient broth. The pH of the nutrient broth containing dye at a concentration of 20 ppm was adjusted to different pH values from 1 to 13 by adding 1 N of NaOH or $H_2SO_4$ to the nutrient broth. The samples were incubated at 37 °C for three dyes after *Pseudomonas aeruginosa* inoculation. After three dye incubations, each culture tube with a specified pH was centrifuged for 20 min at 5000 rpm, and we determined the percent degradation as described above.

2.5.3. Temperature Effect on Brown 703 Degradation

Temperature is a critical factor in the bacterial proliferation process. The rate of bacterial growth increases to its maximum when the temperature is at its optimum. Brown 703 solutions (20 ppm) containing nutrient broth and an inoculum of *Pseudomonas aeruginosa* were incubated at four different temperatures, 25, 38, 45, and 50 °C, for three days. The extent of degradation was noted as described.

2.5.4. Effect of Glucose

Bacteria use glucose as a source of energy for biosynthesis of a number growth factors. To observe the impact of glucose on the degradation of brown 703, different concentrations of glucose, such as 0.1, 0.2, 0.3, 0.4, 0.5, 0.6, 0.7, and 0.8 g/15 mL, were used as an extra carbon source in nutrient broth and incubated for 3 days at 37 °C to degrade the dye. The percent degradation was determined as previously mentioned.

2.5.5. Effect of NaCl (Salt)

Salt is utilized in the textile industry during the dying process because it allows the dye to enter the fabric completely, making the dyeing process more consistent and effortless. The presence of salt affects the efficiency of bacteria in degrading dyes. Different salt concentrations (0.1, 0.2, 0.3, 0.4, 0.5, 0.6, and 0.7 g/15 mL) were employed in the inoculated test tubes and incubated at 37 °C for three days to see the effect of salt on the bacterial degradation of brown 703 and percent degradation was determined as previously mentioned.

2.5.6. Effect of Time

Incubation time is also an important factor for bacterial growth. A 500 mL flask containing brown 703 (20 ppm), nutrient broth, and a *Pseudomonas aeruginosa* inoculum was

cultured for 28 days to determine the influence of time on brown 703 degradation. At the end of every 24 h, 10 mL samples were obtained and centrifuged for 20 min at 5000 rpm to separate the supernatant and pellet. The dye degradation percentage was then determined in the same manner as previously described.

### 2.6. Brown 703 Biodegradation under Optimal Conditions

To achieve maximum degradation and a synergistic effect of different physicochemical parameters on brown 703 degradation by *Pseudomonas aeruginosa*, the optima determined in the above experiments were combined in a single experiment. The remaining experimental conditions were the same as previously stated.

### 2.7. Brown 703 Degradation Metabolite Extraction, Isolation, and Identification

The degraded culture of brown 703 by *Pseudomonas aeruginosa* under optimal conditions was centrifuged for 20 min at 5000 rpm at room temperature. The supernatant from the cell-free culture was utilized to extract the metabolites by adding an equal amount of ethyl acetate and forcefully agitating for 30 min to promote the transfer of metabolites from the supernatant to ethyl acetate. The ethyl acetate was evaporated at 40 °C in a rotary evaporator in order to obtain the solid extract. After the production of the solid extract of the metabolites, a portion of that extract was subjected to column chromatography in order to obtain the purified metabolites on the basis of size, while the remaining portion was used for GC-MS analysis to identify the degraded metabolites of brown 703. To isolate purified metabolites based on size, the crude extract of brown 703 was combined with 70–230 mesh of silica gel to make slurry, which was then dried in the open air to remove any remaining solvent before being analyzed to obtain the purified metabolites. The glass column that was used for column chromatography was 80 cm in height and 3 cm in diameter. The column was filled with silica gel until it reached a height of 45 cm, after which it was washed with 500 mL of n-hexane to remove impurities. When the column was prepared, the slurry was carefully loaded into the column and washed with n-hexane before being eluted with different ratios of ethyl acetate to n-hexane solvent system (1:5, 1:2, 1:1, 2:1, and 5:1) to achieve the desired results. The glass vials were used to collect the column effluents. Fractions based on thin-layer chromatography (TLC) were subjected to a further chromatogram to obtain pure metabolites. During column chromatography, a 5 mL fraction was collected in each glass vial by passing 75 mL of each solvent through the column. Later on, a spectroscopic examination was performed on the purified fraction to identify the components.

#### 2.7.1. Brown 703 FT-IR Analysis

The FT-IR equipment (IL783LB15H) from Perkin Elmer (Waltham, MA, USA) was used to identify the metabolites that were collected from the column during the experiment. The FT-IR analysis was performed in the mid-IR region (600–4000 cm$^{-1}$). For FT-IR analysis, the original dye brown 703 was used as a control.

#### 2.7.2. Brown 703 GC-MS Analysis

Gas chromatography and mass spectrometry (Thermo GC-TRACE ULTRA VER: 5.0, Thermo MS DSQ-II, Thermo Fisher Scientific, Waltham, MA, USA) analyses were performed using helium gas as the carrier. The GC-MS flow rate was kept constant at 1 mL/min. For the first minute, the temperature of the column was kept constant at 40 °C. The temperature was then gradually raised at a rate of 15 °C per min, from 40 to 240 °C. The temperature of the column was maintained constant at 240 °C for 4.0 min, while the temperature of detector was kept constant at 250 °C. The retention times of the metabolites were compared to those reported in the National Institute of Standards and Technology (NIST) library.

## 3. Results and Discussion

### 3.1. Bacterial Strains Isolation and Identification from Dye Contaminated Water

The dye-polluted water samples were serially diluted, and we used the streaking plate method to isolate a single colony of different bacterial strains on nutrient agar. The nutrient agar plate was incubated for 24 h to visualize the bacterial colonies. After 24 h of incubation, different bacterial colonies appeared on the nutrient agar, of which fifteen colonies were further purified by growing on separate nutrient agar plates. Each of the purified bacterial strains was transferred to nutrient broth for further studies. The isolated bacterial strains were used for the degradation or decolorization of brown 703.

To study the efficiency of different isolated bacterial strains for the degradation of dye, the inoculated tubes were then incubated for 3 days as per the procedure described above. Figure 2 shows the degradation percentages of the fifteen isolates that were screened initially. As compared to the other bacterial strains, *Pseudomonas aeruginosa* showed greater degradation efficiency. Within three days of incubation at 37 °C, it showed about 41.36% degradation. To confirm further the identity of the highest dye decolorizing bacteria, DNA isolation protocol was used and subjected to PCR amplification. The PCR product was used for sequencing. After performing the 16s rRNA sequencing, the sequence was run through BLAST (basic local alignment search tool), which showed that the highest decolorizing bacteria was *Pseudomonas aeruginosa*.

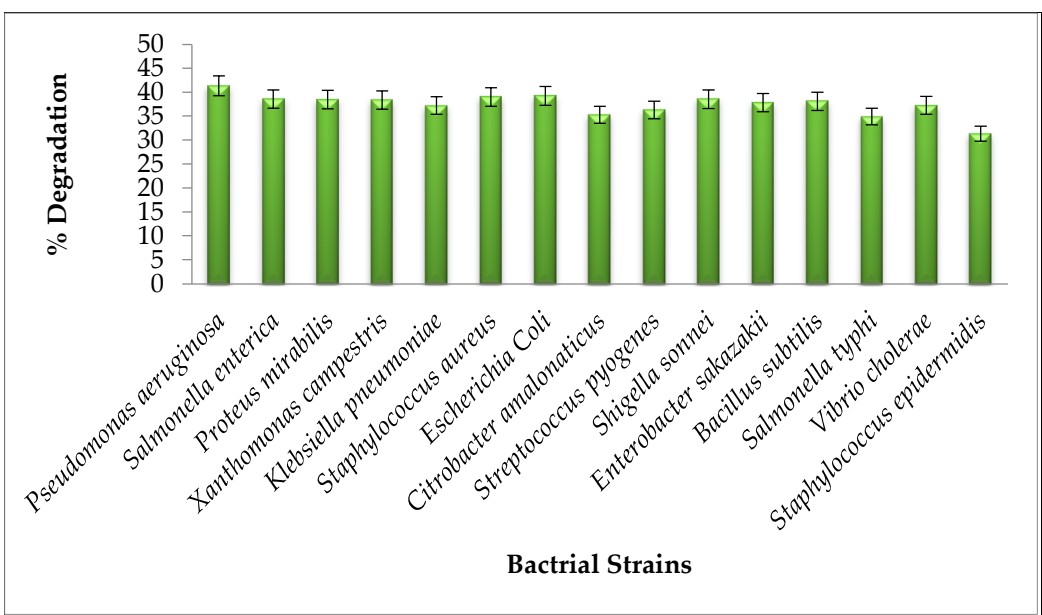

**Figure 2.** Brown 703 dye degradation by different bacterial strains.

### 3.2. Different Parameters and Their Effects on Brown 703 Biodegradation

#### 3.2.1. Effect of Brown 703 Dye Concentration

The degrading efficiency of the *Pseudomonas aeruginosa* was studied at different concentrations (ranging from 20 to 100 ppm) as shown in Figure 3. The highest degradation percentages were found to be 47.65% at 20 ppm, 41.45% at 40 ppm, 34.4% at 60 ppm, 22.01% at 80 ppm, and 13.32% at 100 ppm of brown 703. The increasing concentration has an inhibiting effect on the enzyme reductase, which is responsible for the degradation of the dye [20]. In addition, the presence of the sulphonic group in brown 703 dyes acts as a strong inhibitor of bacterial growth as the number of these groups increases with increased concentration [21]. According to the literature findings, in an industrial effluent treatment system, *Pseudomonas aeruginosa* demonstrates excellent degradation as compared to other bacteria, in spite of the mentioned limiting factors [22].

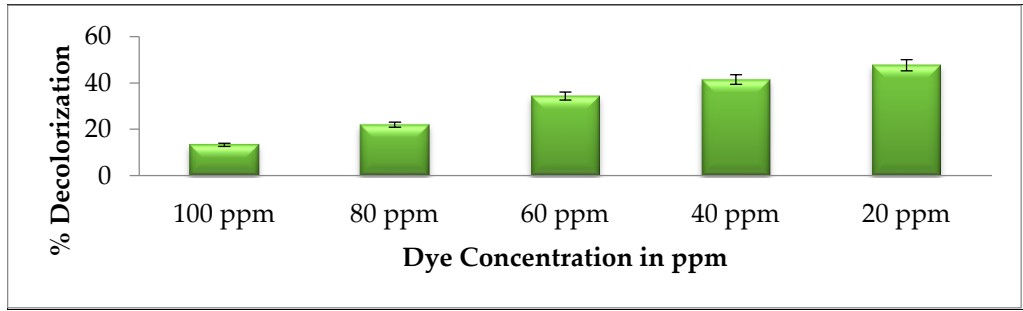

**Figure 3.** Effect of brown 703 dye concentration on % degradation.

### 3.2.2. Effect of pH

pH is an important environmental factor that not only affects bacterial growth but also the activity of the intracellular enzymes involved in degradation processes. The transport of dye molecules into bacterial cells for degradation is also influenced by pH. The effect of pH on *Pseudomonas aeruginosa* was analyzed at different pH levels ranging from 1 to 13, as shown in Figure 4. At a neutral pH, the highest percentage of degradation was observed. The degradation was slower at an acidic pH than at a neutral pH. Protonation of the azo bond makes the bacterium ineffective in interacting with dye, which could be the reason for the decreased inefficiency of *Pseudomonas aeruginosa* at an acidic pH [23]. Textile industrial effluents are basic in nature due to washing, scouring, and mercerizing processes involving the use of basic media [24]. A number of studies have also shown that the degradation efficiency decreases as the pH is raised or lowered from neutral. According to Sheth and Dave [25], the highest degradation of Reactive Red by *Pseudomonas aeruginosa* was found at a neutral pH, which supports our study's findings.

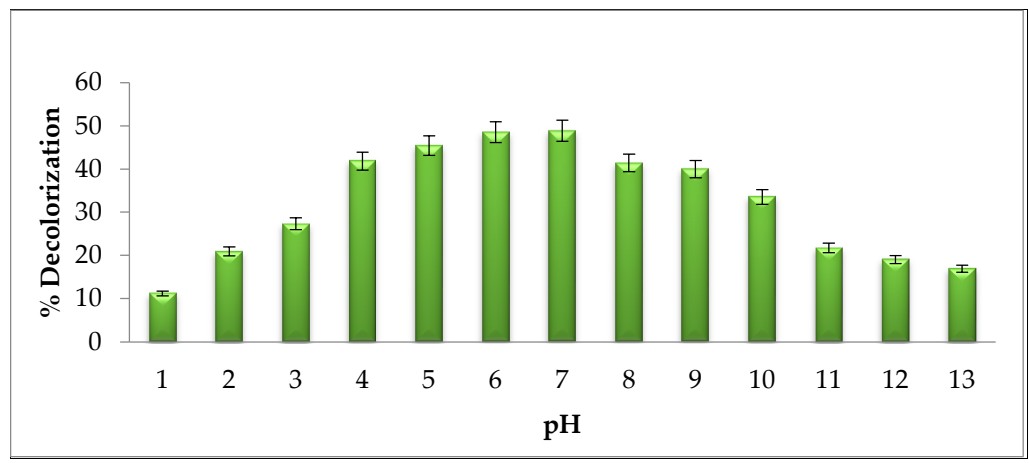

**Figure 4.** Brown 703 % degradation at different pH levels.

### 3.2.3. Effect of Temperature

Temperature affects the activity of bacteria during dye degradation, as shown in Figure 5. The optimum temperature for *Pseudomonas aeruginosa* degradation of brown 703 was found to be 38 °C. The demonstrated degradation efficiencies recorded were 25.85% at 25 °C, 45.49% at 38 °C, 41.02 °C at 45 °C, and 39.98% at 50 °C. The increasing and decreasing of temperatures from their optimum range may result in a decrease in the degradation efficiency due to the decrease in bacterial growth and inactivation of bacteria enzymes as reported previously [26]. The high temperature causes denaturation of bacterial enzymes [27]. According to Kapoor et al. [28], the highest degradation of azo dye degradation by *Pseudomonas aeruginosa* occurred at 37 °C, which supports our research results.

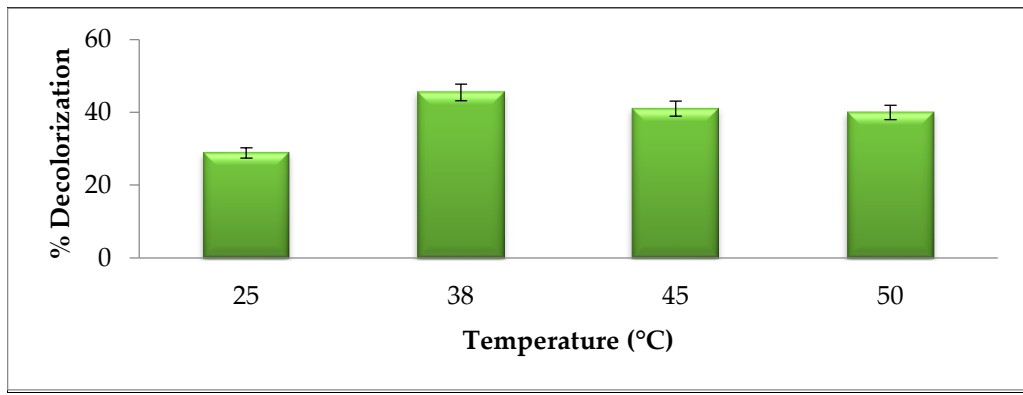

**Figure 5.** Brown 703 % degradation at different temperatures.

### 3.2.4. Effect of Glucose Concentration on Brown 703 Degradation

*Pseudomonas aeruginosa* degradation experiments were conducted in the presence of different glucose concentrations in order to evaluate the effectiveness of the bacteria in the presence of additional carbon support. As we know, for bacteria, glucose acts a source of carbon during their normal growth cycle, providing additional biomass for biodegradation [29]. The highest degradation was noted to be 50.34% at 0.5 g supplementation of the inoculum with glucose, as shown in Figure 6. The degradation efficacy of azo dye decreases below and above 0.5 g/mL due to sugar's catabolic suppression above 0.5 g/mL. Liu et al. [30] reported that the use of glucose results in more efficient degradation of the dye due to its simple uptake and speedy metabolism for the growth of organisms.

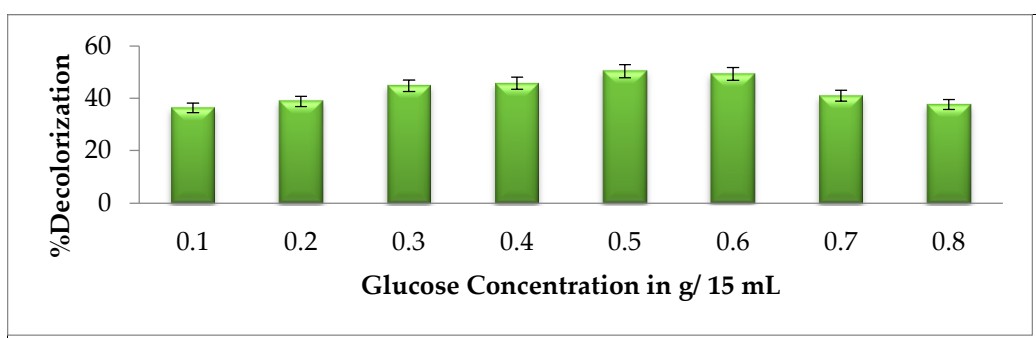

**Figure 6.** Brown 703 % degradation at different glucose concentrations.

### 3.2.5. Effect of NaCl Salt Concentration on Brown 703 Degradation

In response to the high salt usage in the textile industry [31], we investigated the microorganism-degrading capability of brown 703 in the presence of various salt concentrations (0.1 to 0.7 g per inoculum used). The optimum salt concentration was found to be 0.1 g of NaCl supplementation (Figure 7). As the salt concentration increases, the degradation of the dye decreases. A high salt concentration encourages plasmolysis of bacteria cells, which reduces the growth of bacteria and, as a result, causes a reduction in the degradation of azo dye [32].

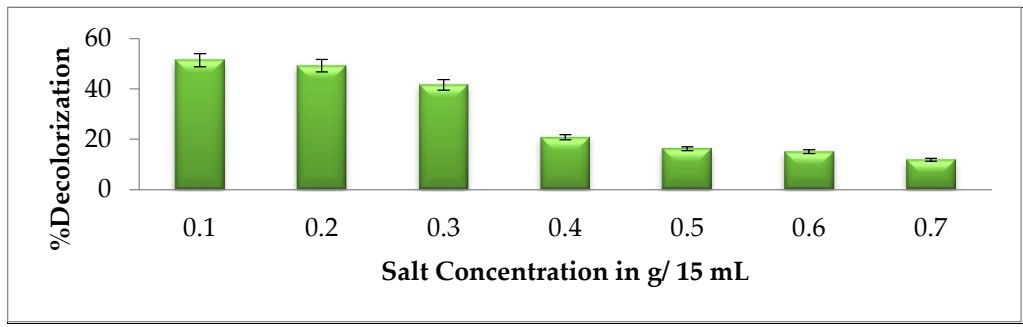

**Figure 7.** Brown 703 % degradation at different salt concentrations.

### 3.2.6. Effect of Time (in Days) on Dye Degradation

Time also plays an important role in the degradation of dye, as bacterial mass proliferation takes time. Figure 8 depicts the effect of time on the degradation of brown 703 by *Pseudomonas aeruginosa*. As time increases, the degradation of dye also increases, to a certain extent. However, after three days, the degradation was not as significant as it was initially. Therefore, three days of incubation was considered to be the ideal period of degradation of brown 703. Initially, the degradation of dye increases with the time interval due to the rapid growth of bacteria in the nutrient broth. After three days, no significant changes were observed due to increase in bacterial biomass and a decrease in nutrients available for bacteria in the medium [33,34].

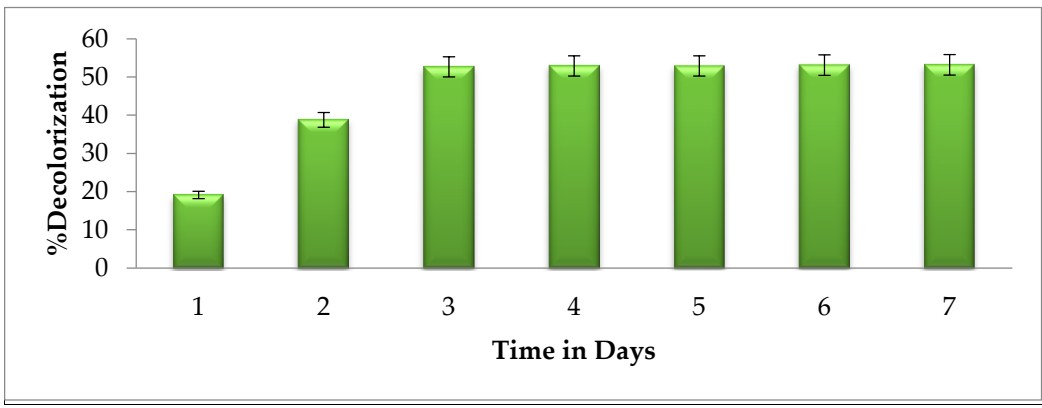

**Figure 8.** The effect of incubation time on the percentage degradation of brown 703 dye.

### 3.3. Biodegradation of Brown 703 at Optimum Conditions

To achieve maximum degradation, the conditions such as temperature, salt concentration, dye concentration, concentration of glucose, pH, and time (in days) were optimized in a series of experiments. After establishing the optima of all of physiochemical conditions for *Pseudomonas aeruginosa*, they were applied in a single experiment for the highest degradation of the selected dye, which further enhanced the degrading ability of the dye to 71.36%.

### 3.4. Characterization of Brown 703 Degraded Metabolites

#### 3.4.1. FT-IR Analysis of Brown 703 Metabolites

The FT-IR spectra of untreated brown 703 dye are shown in Figure 9a. The presence of a peak at 3398 cm$^{-1}$ indicates the presence of amine N–H stretching. N=N stretching is represented by the peak around 1583 cm$^{-1}$, while the peak at 1489 cm$^{-1}$ represents aromatic C=C stretching. The benzene ring connected to the N–H group has peaks ranging from 833 to 745 cm$^{-1}$, while C–H stretching has peaks ranging from 699 to 900 cm$^{-1}$.

When the original dye's FTIR spectrum is compared to the metabolite spectra, significant differences can be seen, which are shown in Figure 9b. Some of the peaks vanished,

while others appeared, showing that the dye was degraded. The majority of bacterial strains have azoreductases, which are enzymes responsible for the breakdown the azo dyes. The peak at 1583 cm⁻¹ disappeared, indicating that the azo linkage was broken by bacterial azoreductase. The peak around 2984 cm⁻¹ represents the =C–H stretching on the benzene ring. The peak at 2892 cm⁻¹ represents the brown 703's saturated C–H stretching. The peak at 1478 cm⁻¹ represents the C–C bond stretching. C–N bond stretches are represented by the peaks at 1235 and 1046 cm⁻¹. Generally, the two spectra (treated and untreated dye) are vastly different from one another, making correlation extremely difficult. The most notable finding is the disappearance of the azo bond peak, which can be confirmed using NMR data of the degraded dye as well.

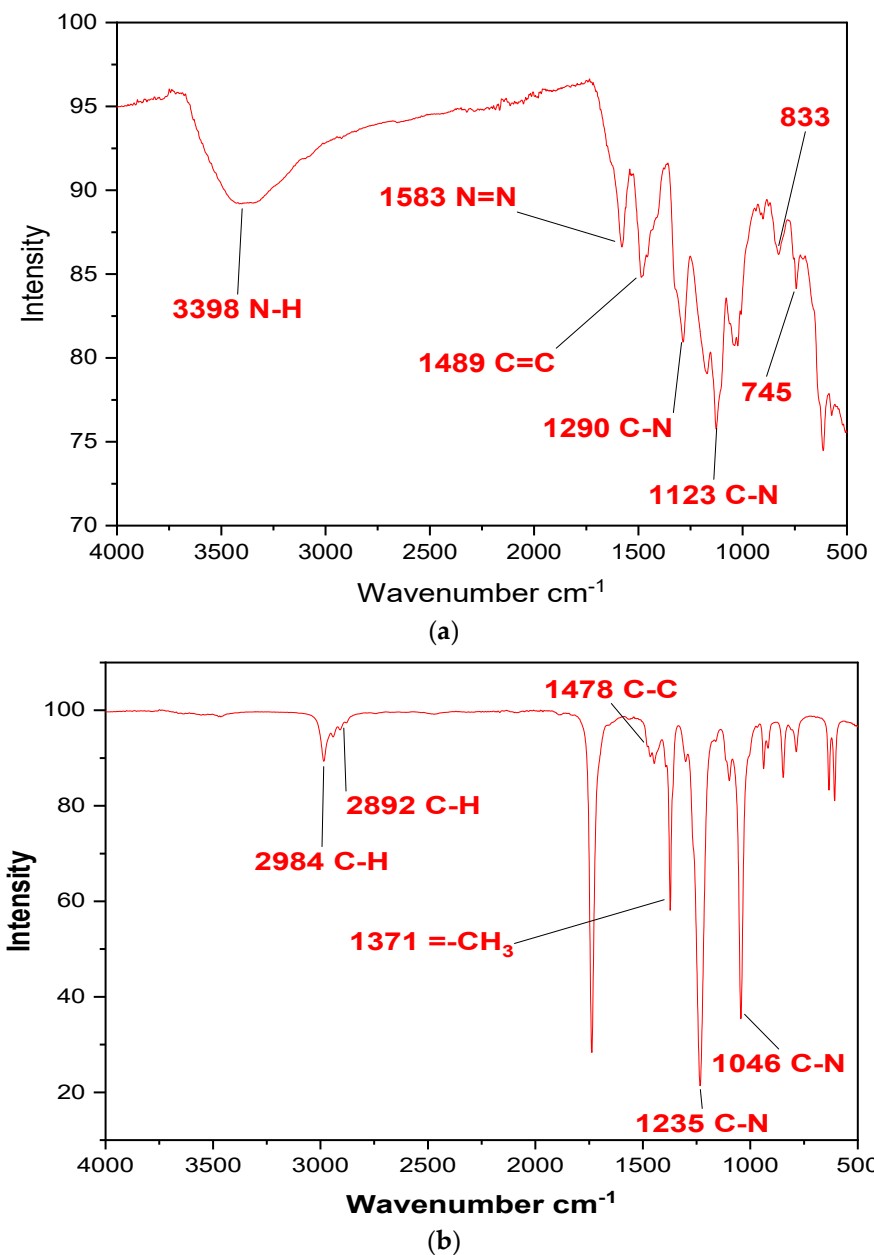

**Figure 9.** (**a**) Brown 703 original dye FT-IR. (**b**) Brown 703 dye FT-IR after *Pseudomonas aeruginosa* degradation.

### 3.4.2. GC-MS Analysis of Brown 703 Metabolites

Figure 10 shows the GC-MS chromatograms of the brown 703 metabolites mixture, while Table 1 includes the relevant compounds identified. The individual compounds' fragmentation patterns along with structures are given in Figure 11. The majority of the compounds are likely to be present in solvents due to the usage of commercial-grade solvents. The compound at RT 1.30 min with a charge to ion mass of 92 *m/z* is related to the dye structure and was identified as toluene, which was further validated using carbon-13 and proton NMR.

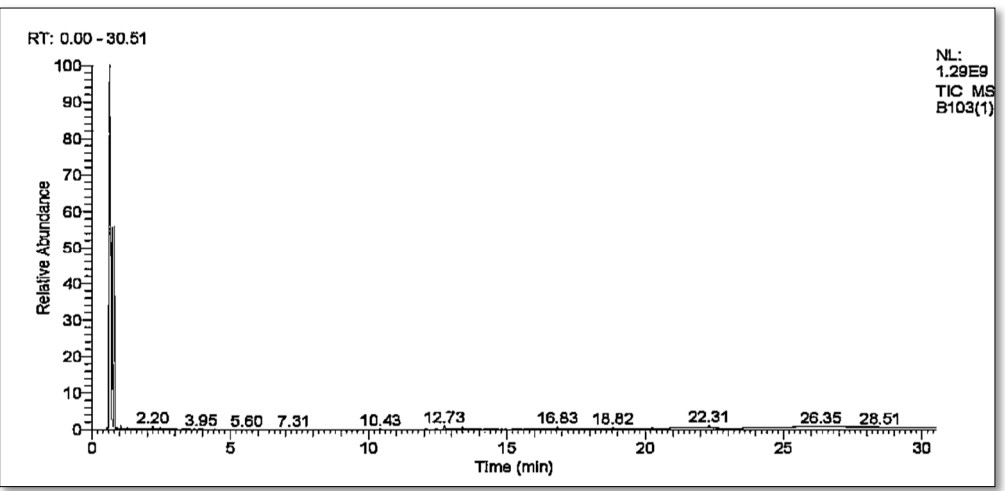

**Figure 10.** Brown 703 dye GC chromatogram after *Pseudomonas aeruginosa* degradation.

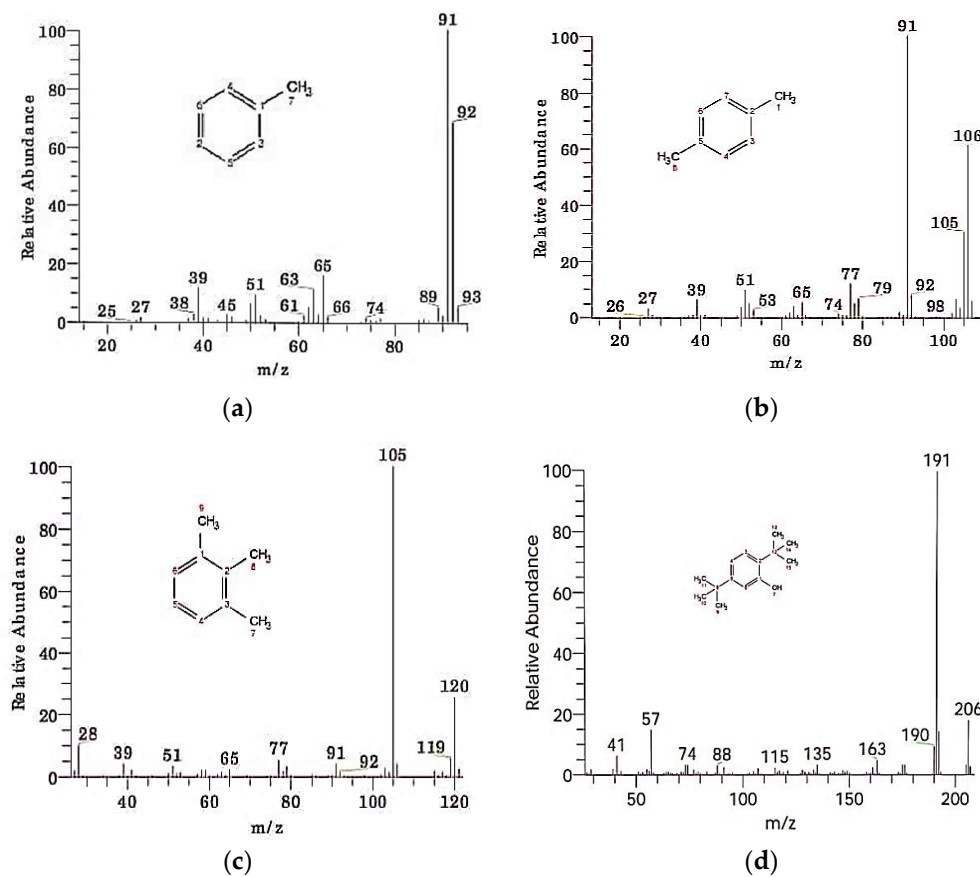

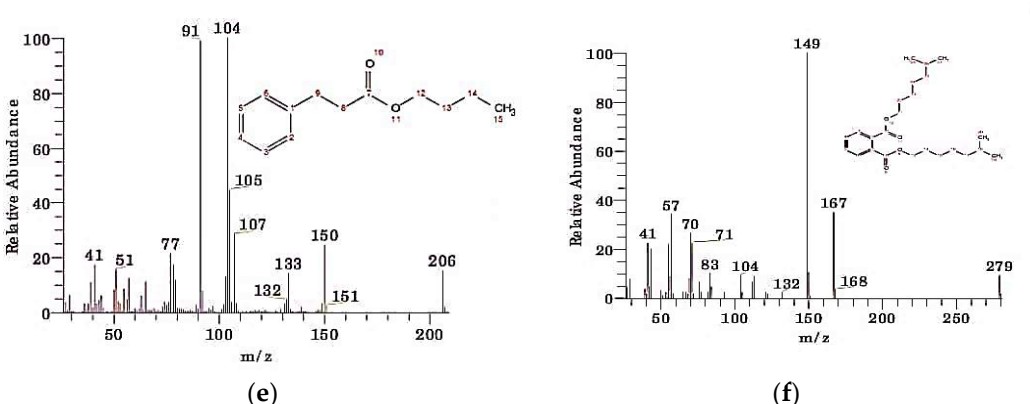

(**e**)                                                                  (**f**)

**Figure 11.** GC/GC-MS chromatograms of brown 703 crude extract. (**a**) Toluene; (**b**) p-xylene; (**c**) benzene, 1,2,3-trimethyl; (**d**) phenol, 2,5-bis(1,1-dimethylethyl); (**e**) benzene propanoic acid, butyl ester, and (**f**) 1,2-benzenedicarboxylic acid, diisooctyl ester.

**Table 1.** Brown 703 GC-MS identified compounds from the degraded mixture.

| S. No. | Compound Name | Retention Time | Peak Area | Chemical Formula | Molecular Weight |
|--------|---------------|----------------|-----------|------------------|------------------|
| 1 | Toluene | 1.30 | 0.26 | $C_7H_8$ | 92 |
| 2 | p-Xylene | 2.18 | 0.39 | $C_8H_{10}$ | 106 |
| 3 | Benzene, 1,2,3-triMethyl | 3.42 | 0.03 | $C_9H_{12}$ | 120 |
| 4 | Phenol, 2,5-bis(1,1-diMethylethyl) | 12.30 | 0.07 | $C_{14}H_{22}O$ | 206 |
| 5 | Benzenepropanoic acid, butyl ester | 12.75 | 0.50 | $C_{13}H_{18}O_2$ | 206 |
| 6 | 1,2-Benzenedicarboxylic acid, diisooctyl ester | 22.31 | 0.51 | $C_{24}H_{38}O_4$ | 390 |

### 3.4.3. Brown 703 Dye NMR Spectra of Metabolites

There was a notable difference between NMR spectra of the original dye (brown 703) and after degradation by the bacterial strain *Pseudomonas aeruginosa*. Figure 12a,b show the [1]H NMR and [13]C NMR results of the original brown 703 dye. After *Pseudomonas aeruginosa* degradation of brown 703 and the isolation of metabolites, the mixture was subjected to column chromatography in order to separate the metabolites based on their sizes/polarities, which were then studied via NMR. Only toluene was validated as a metabolite via NMR analysis out of the detected compounds using GC-MS. Figure 13a shows the 1H NMR spectra of toluene, while Figure 13b shows the [13]C NMR spectra.

[1]H NMR: The peaks at 2.08 δppm were ascribed to the aliphatic-$CH_3$ (3H, s) protons. Methanol, which is utilized as a solvent, was ascribed to the peaks at 3.27, 3.39, and 4.90 δppm. Aromatic protons are thought to be responsible for the peaks that appear at 7.02 and 7.25 δppm.

Brown 703 dye [13]C NMR revealed two types of peaks: aliphatic methyl carbon and aromatic carbon. Aliphatic methyl carbon was ascribed to the peaks at 20.86 δppm, while the peaks at 125.11, 128.22, 129.92 and 136.25 δppm were ascribed to aromatic carbons. The multiple peaks which are present at 47.32, 47.40, 47.62, 47.97, and 48.08 δppm were ascribed to methanol carbons.

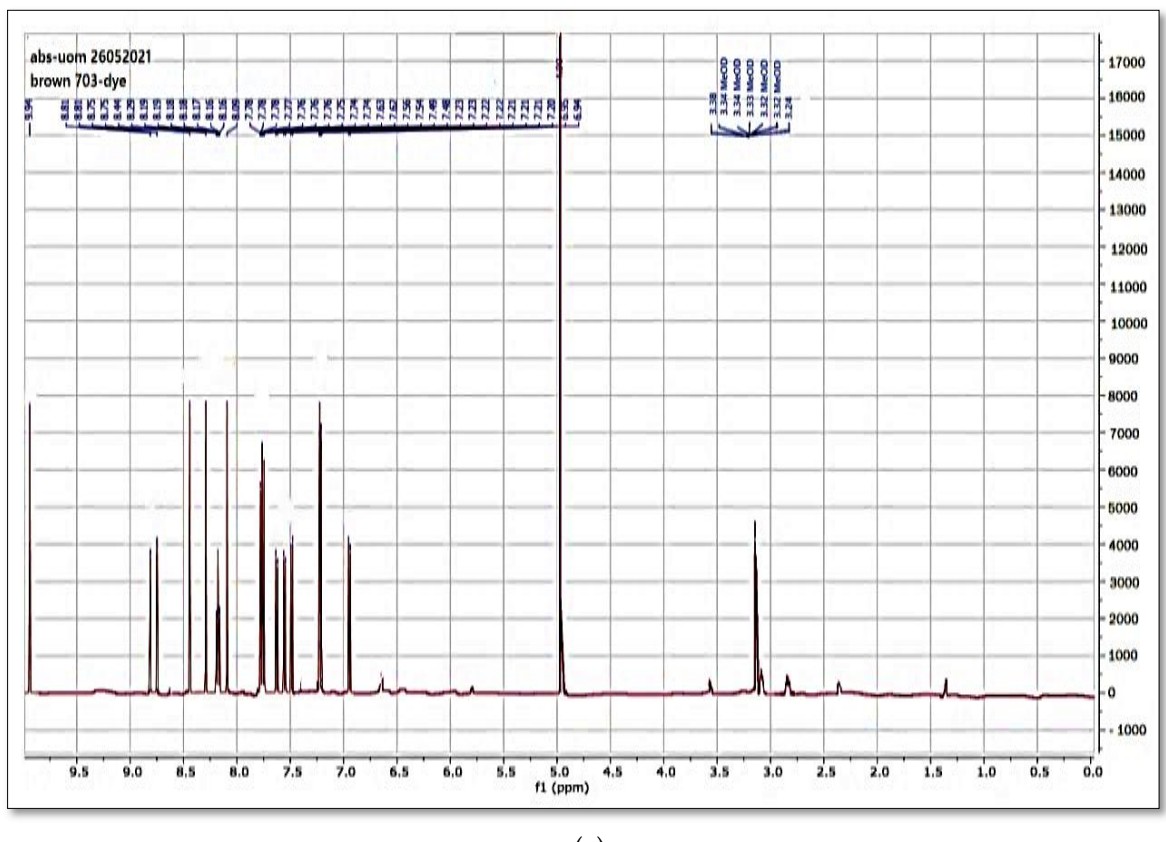

(**a**)

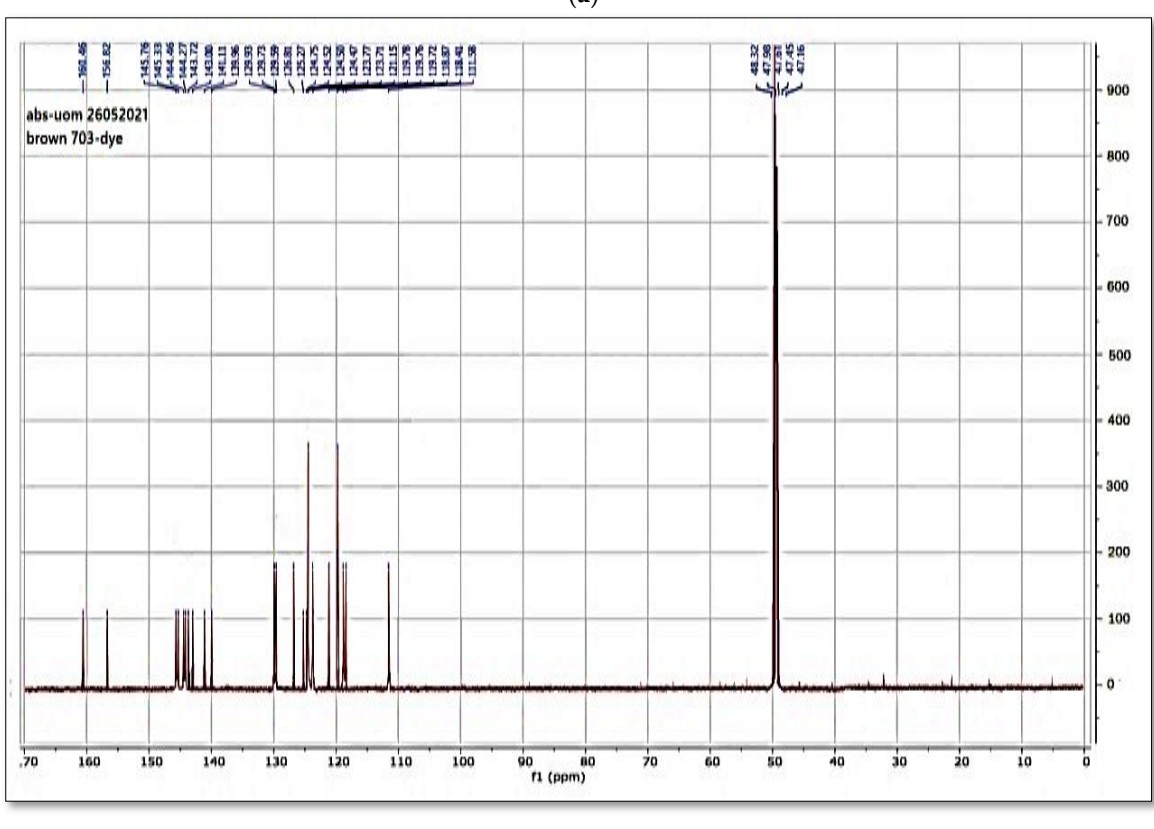

(**b**)

**Figure 12.** (**a**) 1H NMR of original brown 703 dye. (**b**) 13C NMR of original brown 703 dye.

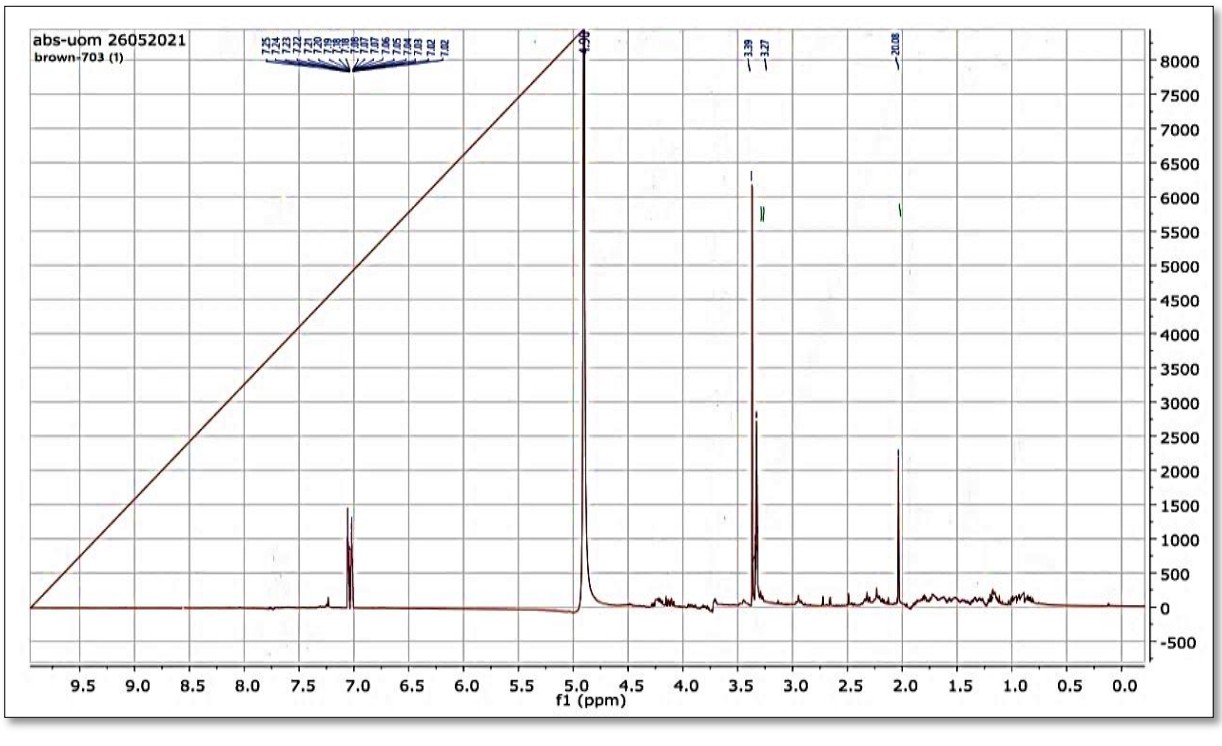

(**a**)

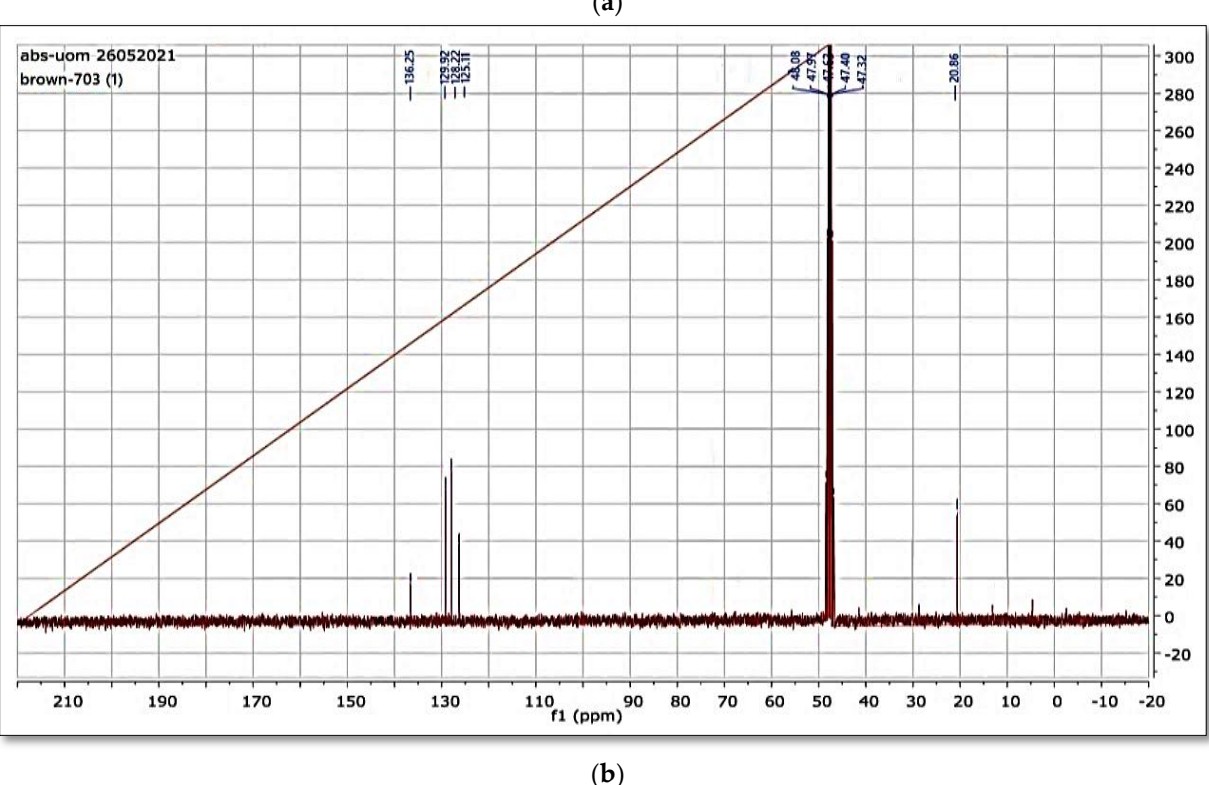

(**b**)

**Figure 13.** (**a**) 1H NMR of brown 703 after degradation by *Pseudomonas aeruginosa.* (**b**) 13C NMR of brown 703 after degradation by *Pseudomonas aeruginosa.*

*3.5. Proposed Mechanism Responsible for the Biodegradation of Brown 703*

Bacteria produce enzymes like azoreductase, peroxidase, and laccase, which are necessary for the breakdown of azo dyes [18,35–37]. The *Pseudomonas aeruginosa* bacteria contained the azoreductase enzyme which breaks the –N=N– bond under aerobic and anaerobic situations. As a consequence, two substituted benzene-based derivatives are

produced via bacterial degradation. Under anaerobic conditions, the sulphonate substituted ring is the most resistant to breakdown [38]. The deamination of the nitro and amino group-substituted rings occurred where the first amino group was converted to a nitro group, and then substituted with a methyl group, resulting in the formation of toluene. Methyl donors such as S-adenosyl-methionine and methyltetrahydrofolate are already present in bacteria and have been associated with benzene ring methylation [39,40]. The suggested mechanism of brown 703 breakdown by *Pseudomonas aeruginosa* is shown below in Scheme 1.

**Scheme 1.** Proposed mechanism for brown 703 degradation by *Pseudomonas aeruginosa.*

### 4. Conclusions

In this study, the azo dye brown 703 was first subjected to fifteen different bacterial strains for degradation which were isolated from the industrial contaminated wastewater. The most efficient bacterial strain in degrading the selected dye was found to be *Pseudomonas aeruginosa*. The best physiochemical conditions for the bacteria to break down the dye were found to be a 20 ppm dye concentration, a neutral pH, a temperature of 38 °C, and a time of 72 h. In a single experiment that integrated the optimization of physio-chemical conditions, a 71.36% degradation of the selected dye was achieved. The metabolites of final degradation were subjected to silica gel column chromatography to separate on the basis of size/polarity. The dye metabolites which were produced during bacterial degradation were characterized by using FT-IR, GC-MS and NMR spectroscopy. The spectroscopic data confirmed the presence of toluene, which was also validated by GC-MS. Because of the activity of azoreductase, the dye was cleaved, and the subsequent

deamination and methylation resulted in the formation of toluene. It can be concluded from the above results that *Pseudomonas aeruginosa* could be effectively used as a potent strain for wastewater treatment containing azo dyes.

**Author Contributions:** Conceptualization, A.U.K., M.Z. and M.I.; methodology, A.U.K., M.I. and M.Z.; software, A.U.K.; M.U.R. and M.Z.; validation, M.U.R., M.I. and E.A.A.; formal analysis, M.N.U., D.Z. and R.U.; investigation, M.N.U., R.U., E.A.A. and D.Z.; resources, M.Z.; data curation; A.U.K.; M.U.R. and M.Z.; writing original draft preparation: A.U.K., M.Z. and M.I.; project administration: M.Z. and M.U.R.; funding acquisition, R.U. and E.A.A. All authors have read and agreed to the published version of the manuscript.

**Funding:** Researchers Supporting Project number (RSP2023R110) King Saud University, Riyadh, Saudi Arabia

**Institutional Review Board Statement:** Not applicable.

**Informed Consent Statement:** Not applicable.

**Data Availability Statement:** The data associated with this manuscript has been fully presented in this paper.

**Acknowledgments:** The authors extend their appreciation to the researchers supporting Project number (RSP2023R110) King Saud University, Riyadh, Saudi Arabia, for their financial support.

**Conflicts of Interest:** The authors declare no competing interest. The funders did not participate in the study's design, data collection, analysis, or interpretation, manuscript writing, or decision to publish the findings/results.

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
