# Peer review of "Bioremediation of Azo Dye Brown 703 by Pseudomonas aeruginosa: An Effective Treatment Technique for Dye-Polluted Wastewater"

_2036-7481, doi:10.3390/microbiolres14030070_

Round 1

Reviewer 1 Report

This study focuses on the degradation of the dye brown 703 using Pseudomonas aeruginosa, a bacterial strain isolated from textile wastewater dumping sites in Mingora, Swat.

The researchers conducted optimization experiments on the nutrient broth medium, subjecting it to various environmental conditions and nutritional sources. They found that under micro-aerophilic circumstances, the maximum decolorization and degradation of the dye occurred at a concentration of 20 ppm within three days of incubation at neutral pH and 38 °C. Out of the fifteen bacterial strains initially isolated from textile effluent, Pseudomonas aeruginosa exhibited the highest degradation rate of approximately 71% under optimum conditions. The FT-IR analysis revealed the appearance of new peaks after the degradation of brown 703, indicating that Pseudomonas aeruginosa was responsible for the dye degradation. Additionally, the GC-MS study helped identify the degraded azo dye compounds, providing insights into the degradation process.

The study demonstrates a potential approach for the biodegradation of brown 703 dye, and findings indicate that Pseudomonas aeruginosa is effective in removing color from textile wastewater, suggesting its potential industrial use in textile wastewater treatment.

Overall, this study highlights the importance of wastewater treatment. However, there are a few issues to be addressed.

Having in mind the error of 2-5%, all numbers should be without decimal numbers in Figs 2-8.

The sentence: The results show that the increase in dye concentration is accompanied by a decrease in dye degradation percent due to the toxic nature of dye, which inhibits bacterial growth.

must be removed, this is expected, and it is not reasonable to underline this fact. This result will be seen even if the tested dye isn't toxic.

Fig 9 a and b should be given as tiff files, not as distorted non-vector images.

Fig. 11 should be plotted in Origin or similar vector drawing tools for diagrams; the image is not acceptable as is.

The most important is to highlight what is novel in this work since a number of papers deal with this kind of bioremediation.

Minor editing of English language required

Author Response

Reviewer 1

Comments and Suggestions for Authors

This study focuses on the degradation of the dye brown 703 using Pseudomonas aeruginosa, a bacterial strain isolated from textile wastewater dumping sites in Mingora, Swat.

The researchers conducted optimization experiments on the nutrient broth medium, subjecting it to various environmental conditions and nutritional sources. They found that under micro-aerophilic circumstances, the maximum decolorization and degradation of the dye occurred at a concentration of 20 ppm within three days of incubation at neutral pH and 38 °C. Out of the fifteen bacterial strains initially isolated from textile effluent, Pseudomonas aeruginosa exhibited the highest degradation rate of approximately 71% under optimum conditions. The FT-IR analysis revealed the appearance of new peaks after the degradation of brown 703, indicating that Pseudomonas aeruginosa was responsible for the dye degradation. Additionally, the GC-MS study helped identify the degraded azo dye compounds, providing insights into the degradation process.

The study demonstrates a potential approach for the biodegradation of brown 703 dye, and findings indicate that Pseudomonas aeruginosa is effective in removing color from textile wastewater, suggesting its potential industrial use in textile wastewater treatment.

Overall, this study highlights the importance of wastewater treatment. However, there are a few issues to be addressed.

  • Worthy reviewer thank you very much for reviewing and highliting the issues in our paper, by removing these issues quality of paper was improved.

Comment: Having in mind the error of 2-5%, all numbers should be without decimal numbers in Figs 2-8.

Response: Worthy reviewer, the error bars have correct accordingly by removing all the numbers from Figs 2-8.

Comment: The sentence: The results show that the increase in dye concentration is accompanied by a decrease in dye degradation percent due to the toxic nature of dye, which inhibits bacterial growth.must be removed, this is expected, and it is not reasonable to underline this fact. This result will be seen even if the tested dye isn't toxic.

Response: The sentence has been removed accordingly. Thank you for the worthy suggestion.  

Comment: Fig 9 a and b should be given as tiff files, not as distorted non-vector images.

Response: Worthy reviewer these figures have been given as tiff files.

Comment: Fig. 11 should be plotted in Origin or similar vector drawing tools for diagrams; the image is not acceptable as is.

Response: Worth reviewer, in Pakistan, the research instruments are limited in numbers and on payment we only get results in PDF and printed form not in soft. Then on sniping tool we convert them to picture. We do not have data of the picture in editable format to make such changes.

Comment: The most important is to highlight what is novel in this work since a number of papers deal with this kind of bioremediation.

Response: Worthy reviewer, though a number of studies highlights the degradation abilities of the microbes but none of the researchers have made the attempt to isolate the metabolites in pure form to get an insight into the degradation process. Based on the isolated metabolites a mechanism of degaradation has been propsed. The statement was accordingly added to introduction section as well.

Reviewer 2 Report

The article presents the results of research on the potential using of Pseudomonas aeruginosa  in the treatment of colored wastewater, especially azo dye Brown 703.

A novelty in the presented article is the indication of the mechanism of bacterial degradation of the dye 703 by Pseudomonos aeruginosa.

The authors explain the gaps in the knowledge. However, there were several inaccuracies in the article about this mechanism. The manuscript need also some language correction. My comments are listed below:

Abstract: please find other words to describe "living things". Double-check the use of the word "things" throughout the article for the correct spelling. Try to find a synonym for this word.

“azo dye brown 703 was investigated using Pseudomonas aeruginosa” –confusing

Keywords: try to use other keywords (not that are using in the article) to make the article more visibility

Introduction:

L82: wastewater

L99: Could you provide literature data on the toxicity of the resulting metabolites?

L105: the specific objective of this research was…

L112: remove a drop from the sentence

Methods:

-          supplement the methodology with information whether it is your own developed methodology or a methodology based on someone else's earlier work

-          how was the bacteria distinguished and confirmed to be Pseudomonas aeruginosa?

L121-129: Is it your own method for isolation of bacterial strains?

L128: fifteen – use the number 15

L140: please change pallets onto pellets – check through whole article

L140: 703dye –  separately

L233: cm-1

L256: The supernatant was used for percent degradation – very confusing

L262: explain the abbreviation BLAST

L274: Azam et al., - remove “,”

Figure 3: why there is little difference in degradation efficiency for 40 ppm and 20 ppm. And why is the effectiveness almost two times lower for the dose of 80 ppm compared to 40 ppm?

3.2.4. Why do you calculate by using gram per 15ml. Present the calculated value in g/ml

L315: it is mentioned the value 0.5 g/ml and in the figure 6 the unit is gram/15ml?

Figure 6: the author did not explain why the efficiency of degradation decreases despite increasing the dose of glucose.

L341-342: did the author consider adding more nutrient to extend the time of bacterial growth and contact with the dye

Figure 8: the effect of time (days) on brown 703 % degradation - confusing

L420: our research experiment was carried out in both anaerobic and aerobic conditions – where it was mentioned in the methods part?

Author Response

Reviewer 2

Comments and Suggestions for Authors

 The article presents the results of research on the potential using of Pseudomonas aeruginosa  in the treatment of colored wastewater, especially azo dye Brown 703.

A novelty in the presented article is the indication of the mechanism of bacterial degradation of the dye 703 by Pseudomonos aeruginosa.

The authors explain the gaps in the knowledge. However, there were several inaccuracies in the article about this mechanism. The manuscript need also some language correction. My comments are listed below:

  • Thank you worthy reviewer for the positive input. The language of the paper was accordingly improved including the mechanism section.

Comment: Abstract: please find other words to describe "living things". Double-check the use of the word "things" throughout the article for the correct spelling. Try to find a synonym for this word.

Response: These words have been replaced with other appropriate words.  The spelling of the word "things" has been checked throughout  the manuscript. The paper was extensively edited by native speaker, hope not it will be ok.

Comment: “azo dye brown 703 was investigated using Pseudomonas aeruginosa” –confusing

Response: Worthy reviewer: the sentence was rephrased to (The textile azo dye brown 703 was degraded using Pseudomonas aeruginosa). The paper was extensively edited by native speaker, hope not it will be ok.

Comment: Keywords: try to use other keywords (not that are using in the article) to make the article more visibility

Response:Thank you dear refree. Keywords have been changed accordingly.

Introduction:

Comment: L82: wastewater

Response:Worthy reviewer: this mistake has been corrected.

Comment: L99: Could you provide literature data on the toxicity of the resulting metabolites?

Response: In another study we have conducted phytotoxicity study and the resultant metabolites are less toxic than parent dye. As per literature reports these metabolites are less toxic than parent dye .

Comment: L105: the specific objective of this research was…

Response: Worthy reviewer: correct accordingly

Comment: L112: remove a drop from the sentence

Response: Worthy reviewer: this mistake has been  corrected accordingly

Comment: Methods: supplement the methodology with information whether it is your own developed methodology or a methodology based on someone else's earlier work

-          how was the bacteria distinguished and confirmed to be Pseudomonas aeruginosa?

Response: To confirm to the highest dye decolorization bacteria,we  first isolate the bacterial DNA using bacteria DNA isolation protocol and subjected it to PCR amplification. The PCR product was used for sequencing. When 16s rRNA sequencing was done of high-degradation bacterial strain, the sequence was then run through BLAST, which showed that the highest decolorization bacteria was Pseudomonas aeruginosa.

Comment: L121-129: Is it your own method for isolation of bacterial strains?

Response: Worthy reviewer:  we have used Streck and Pour plate methods to isolate and purify bacterial strains from dye-contaminated water.

Comment: L128: fifteen – use the number 15

Response: Worthy reviewer: This  mistake has been corrected accordingly

L140: please change pallets onto pellets – check through whole article

Response: This  mistake has been corrected accordingly.Thank you

Comment: L140: 703dye –  separately

Response: Worthy reviewer; corrected accordingly

Comment: L233: cm-1

Response: Worthy reviewer; corrected accordingly

Comment: L256: The supernatant was used for percent degradation – very confusing

Response: This sentence was rephrased as; The supernatant of the sample was used for the percent degradation of dye by different bacterial strains.

Comment: L262: explain the abbreviation BLAST

Response: The abbreviation BLAST has been explained. Thank you

Comment: L274: Azam et al., - remove “,”

Response:Dear reviever: Citation in text has been changed accordingly. Azam et al has been removed. .  

Comment: Figure 3: why there is little difference in degradation efficiency for 40 ppm and 20 ppm. And why is the effectiveness almost two times lower for the dose of 80 ppm compared to 40 ppm?

Response: Degradation efficiency of bacteria increase with increase in dye concentration upto certain limit. In our study 20 ppm is optimum concentration for better degradation.   At  high  concentration of dye have toxic effect on bacterial growth. Zhuang et al. 2020 reported toxicity and the blockage of enzyme active sites with high concentrations of dye. High concentration of dye need more biomass of bacteria. 40 ppm was not strongly toxic as compared to other concentrations tested.

  • Saratale, R.G.; Saratale, G.D.; Chang, J.S.; Govindwar, S.P. Bacterial decolorization and degradation of azo dyes: A review. J. Taiwan Inst. Chem. Eng. 2011, 42, 138–157
  • Zhuang, M.; Sanganyado, E.; Zhang, X.; Xu, L.; Zhu, J.; Liu, W.; Song, H. Azo dye degrading bacteria tolerant to extreme conditions inhabit near shore ecosystems: Optimization and degradation pathways. J. Environ. Manag. 2020, 261, 110222–110231

Comment: 3.2.4. Why do you calculate by using gram per 15ml. Present the calculated value in g/ml

Response: Worthy reviewer; you are right however in our presented figure the results are based on this amount then it will confusing for readers, though we have omitted 15 ml from revised paper.

Comment: L315: it is mentioned the value 0.5 g/ml and in the figure 6 the unit is gram/15ml?

Response:Dear referee; this is typo mistake. It is 0.5 g/15ml, however to avoid confusion we have used the word inoculum.

Comment: Figure 6: the author did not explain why the efficiency of degradation decreases despite increasing the dose of glucose.

Response: Worthy reviewer: Some time high concentration of glucose have negative effect on metabolism which is known as “sugar catabolic suppression”. Due to sugar catabolic suppression above 0.5 g/mL the decrease in bacterial degradation of dye occurs. This detail has been added into revised paper as well.

Comment: L341-342: did the author consider adding more nutrients to extend the time of bacterial growth and contact with the dye.

Response: Worthy reviewer: In our research study, we have used the fixed nutrients method for bacterial degradation of dye. If we use a continuous nutrient supply method for bacteria, bacteria potential may incresese to degrade the dye completely.

Comment: Figure 8: the effect of time (days) on brown 703 % degradation – confusing

Response: Figure 8: title was changed to: The effect of incubation time on the percentage degradation of brown 703 dye

Comment: L420: our research experiment was carried out in both anaerobic and aerobic conditions – where it was mentioned in the methods part?

 Response: Worthy reviewer: Our research used the bacterial strain Pseudomonas aeruginosa, which is aerobic–facultatively anaerobic. It means that the bacteria produced the enzyme in both conditions, such as anaerobic and aerobic. However, as not mentioned in experimental section therefore to avoid confusion the sentence was removed.

Round 2

Reviewer 1 Report

Authors have carefully reviewed the comments and adequately responded to the raised issues which warrants the publication in present form.

Minor style changes can be made.